# Advancements in Induction Motor Fault Diagnosis and Condition Monitoring: A Comprehensive Review

**DOI:** 10.3390/s25195942

**Published:** 2025-09-23

**Authors:** Kamal Hamani, Martin Kuchar, Marek Kubatko, Stepan Kirschner

**Affiliations:** Department of Applied Electronics, Faculty of Electrical Engineering and Computer Science, VSB-Technical University of Ostrava, 17. listopadu 2172/15, 708 00 Ostrava, Czech Republic; kamal.hamani.st@vsb.cz (K.H.); marek.kubatko@vsb.cz (M.K.); stepan.kirschner@vsb.cz (S.K.)

**Keywords:** induction motor, diagnosis, machine learning, condition monitoring, signal processing

## Abstract

Induction motors (IMs) are the backbone of modern industry. Despite their robustness and reliability, they are prone to a range of problems that can result in periods of inactivity, diminished operational efficiency, and potential safety risks. Rapid identification and assessment of faults is important to maintain efficient motors operation and avoid serious malfunctions. The paper offers a comprehensive analysis of the existing body of knowledge in IMs’ faults detection, highlighting areas of deficiency and obstacles. Our review is built according to the IMs diagnosis process, presenting for each step of this process several approaches. Finally, we discuss the effectiveness of each fault classification approach in addressing data-driven challenges such as high-dimensionality, class imbalance, nonlinearity, noise, and overfitting. This paper highlights the rising transition to data-driven strategies, with deep learning increasingly taking center stage in tackling the complex challenges of fault diagnosis. It underscores the significant impact of these advancements on the field, actively facilitating future research into intelligent, real-time condition monitoring systems.

## 1. Introduction

Induction motors (IMs) play a critical role in modern industries, and their malfunction can significantly disrupt operational productivity and economic performance [1]. Fault detection (FD) is crucial for ensuring the continuous operation of vital machinery in industrial production lines, thereby conserving both time and financial resources [2]. Condition monitoring (CM) methods detect equipment abnormalities early, preventing them from escalating into critical failures and enabling timely, proactive maintenance interventions [3,4]. This strategy offers various benefits [5]:Reduced total maintenance costs.Optimized spare parts inventory.Customized proactive maintenance based on equipment needs.Improved safety.

Figure 1 illustrates the predictive maintenance (PM) strategy within the framework of the P–F curve [6], a widely recognized concept in industrial maintenance. This curve represents the health condition of a machine over time, starting from its peak performance to functional failure. The “P” point marks the moment a potential failure is first detected, while the “F” point represents the point of functional failure. Between these two points lies a crucial window for predictive maintenance actions, during which timely detection and intervention can prevent the fault from progressing to a catastrophic breakdown. PM systems leverage real-time data from sensors and diagnostic tools to detect these early signs of failure, allowing corrective measures before severe consequences arise. In contrast to classical methods, which have relied explicitly on historical data, CM uses sensors to provide meaningful information on the health of various devices. These sensors collect data to monitor crucial operating parameters such as vibrations, sound anomalies, airflow, and current. The different types of PM take CM to the next level [7]. Building on these sensor data, PM uses advanced time–frequency analytics and artificial intelligence (AI) to predict machine failures before they occur. CM employing a data-driven approach follows a sequential procedure, commencing with data acquisition, followed by data processing and cleansing, culminating in fault isolation and signature characterization. Upon completion of this process, a predictive model is generated, which can subsequently be deployed for real-time CM applications.

Figure 2 presents a comprehensive workflow of the Condition-Based Monitoring (CBM) process specifically tailored for induction motors. This diagram outlines the sequential steps involved in fault detection and diagnosis, beginning with data acquisition and preprocessing, followed by feature extraction through signal processing techniques, and culminating in fault classification using machine learning algorithms (MLAs). It also highlights the iterative nature of this process, where feedback from fault detection outcomes is used to refine and improve predictive models over time. This structured approach ensures systematic monitoring of IMs’ health status, starting from initial raw data collection to actionable diagnostic conclusions. Each stage of the process offers opportunities to mitigate challenges such as high-dimensional data, class imbalance, nonlinearity, and noise. The figure also emphasizes the integration of both traditional signal analysis methods and advanced machine learning (ML) techniques, reflecting the current trend toward intelligent, data-driven CM systems.

The review is a comprehensive collection of references related to (FD), (CM), and Predictive Maintenance (PM) in IMs. It builds on the structure depicted in Figure 2, examining in detail each step of the diagnosis process, surveying existing methodologies, identifying limitations, and discussing how emerging AI-driven techniques can address prevailing challenges in IMs diagnostics. The paper is organized as follows. Section 2 reports the various failures that affect the proper operation of IMs and presents their categorization. Section 3 reviews the different strategies employed for data acquisition, detailing the nature and characteristics of the collected signals. To address the challenges of noise and irrelevant information in the datasets, Section 4 discusses fault signature extraction and signal processing techniques. Section 5 presents the diverse methods applied in the data classification process. The comparative advantages of these approaches, depending on specific fault scenarios, are analyzed in Section 6. Finally, Section 7 concludes the review and outlines the key findings.

## 2. Induction Motor Faults

Due to harsh operating conditions, IMs are not free of damage that affects the stator, rotor, bearings, etc. The most frequent induction motor defects are bearing failures, representing (from 40% to 50% ) of all other fault occurrences. The IEEE reports that 42% of faults in induction motors (IMs) are attributed to bearing defects, 28% to stator defects, 8% to rotor defects, and 22% to other types of faults. Conversely, the Electric Power Research Institute (EPRI) estimates the likelihood of bearing faults at 41%, stator faults at 36%, rotor faults at 9%, and other faults at 14% [8,9].

Different research studies are being conducted to study various types of motor failure in multiple research settings [9,10]. Some methods are based on the analysis of electrical parameters, and others on non-electrical values as vibration, acoustic, thermal, etc., to detect IM faults like motor eccentricity, and machine insulation faults [11]. A comprehensive grasp of the physical phenomena is essential for creating models tailored to each fault mode, which must also accommodate various operating conditions. Consequently, the resulting programs that generate the diagnostic indices are challenging to implement [7]. Fault categorization in induction motors (IMs) plays a critical role in enabling rapid fault detection and assessing the severity of failures. By classifying the faults accurately, maintenance teams can respond appropriately and in a timely manner. These faults, which can significantly reduce efficiency or even cause catastrophic failures, are broadly classified into three categories: (i) electrical faults (e.g., stator winding failures), (ii) mechanical faults (e.g., bearing defects, rotor bar breakages), and (iii) environmental or operational faults (e.g., misalignment, unbalanced loads). Such structured classification not only supports early diagnosis but also helps in selecting appropriate monitoring techniques and intervention strategies. In this review, particular emphasis is placed on internal mechanical faults and the emerging role of AI-based detection methods.

### 2.1. The Induction Motor Faults Classification

Operators of IMs are under continual pressure to reduce maintenance costs and prevent unscheduled motor downtime. To reduce downtime and also for reliable operation, early detection of motor faults is highly demanding. For this, fault diagnosis of induction motors has become a burning topic for electrical technologists over the last two decades [2]. Typically, the current signature or/and vibration signature is analyzed, and useful information is provided to the IM operators about the health condition of the motor [12]. Although the induction motor is reliable, rugged, and almost unbreakable, it is susceptible to different types of faults. The effects of such faults in induction motors include unbalanced stator currents and voltages, torque oscillations, reduction in efficiency and torque, overheating, and excessive vibration [9]. In addition, these faults can increase the magnitude of certain harmonic components of currents and voltages. IMs’ performance may be affected by the following types of faults:**Electrical Faults:** Faults in this category include unbalanced supply voltage or current, single phasing, under- or overvoltage of current, reverse phase sequence, earth fault, overload, inter-turn short-circuit fault, and crawling.**Mechanical Faults:** Faults in this category include a fractured rotor bar, an imbalanced mass, an eccentric air gap, damaged bearings, a failed rotor coil, and a failed stator coil.**Environmental Faults:** The performance of an IM could be affected by ambient temperature and external moisture. The performance of the equipment can be affected by vibrations caused by several factors, such as defects in the installation and foundation defects [13].

Other types of IM faults are listed in detail in Table 1.

Faults in induction motors have been identified through various signal processing and pattern recognition methods, including Fast Fourier Transform (FFT), wavelet transform, and motor current signature analysis (MCSA). Recently, knowledge-based techniques such as k-Nearest Neighbor (KNN), support vector machines (SVMs), and artificial neural networks (ANNs) have gained traction. A nonlinear feature extraction technique known as kernel independent component analysis is proposed, which utilizes the independent component analysis procedure along with the kernel trick to nonlinearly map Gaussian chirplet distributions into a feature space. This approach is implemented in self-organizing maps to categorize the faults of an induction motor [9].

By organizing faults in this manner, we emphasize the multifaceted nature of IMs failures, showing that effective monitoring systems must be capable of identifying both electrical and mechanical anomalies, as well as environmental influences. The subsequent sections of this paper will build upon this classification, detailing how different CM and FD techniques target these specific fault categories and how recent advances in ML and signal processing contribute to improving detection accuracy.

### 2.2. The Common Fault in Induction Motor

Faults in induction motors can be classified in several ways. In review, we choose to list hereafter the common encounter failures: (1) broken rotor bar fault, (2) unbalanced rotational mass, (3) stabilization fault, (4) crumple, (5) bearing fault. Determining the initial failure of an induction motor can be challenging when numerous defects arise simultaneously [9]. The consequences of such a fault in an induction motor are uneven stator currents and voltages, torque oscillations, decreased efficiency, torque, overheating, and excessive vibration [32]. Furthermore, these motor flaws have the potential to amplify the magnitude of specific harmonic components in electrical currents and voltages. Any of the flaws can influence the performance of an induction motor.

## 3. Data Acquisition Strategies for Fault Diagnosis in IMs

Data acquisition for the diagnosis of IMs can be processed in different ways. However, in this review, we classify it into two main approaches: empirical and modeling. Data acquisition for IMs diagnosis involves collecting various electrical and mechanical parameters from the motor during operation. Some of the key parameters that are typically monitored and collected are listed in Table 2 [33]. By analyzing and processing the collected data using techniques such as traditional signal processing or data-driven methods, different faults and abnormalities in IMs can be diagnosed, enabling predictive maintenance (PM) and improved reliability.

### 3.1. Real Data Collection for Fault Detection in IMs

Real data collection plays a crucial role in developing accurate and generalizable fault detection systems for IMs. Experimental datasets are typically obtained through controlled tests that operate the machines under a variety of conditions including varying loads, speeds, and fault scenarios such as bearing defects, stator winding faults, broken rotor bars, and eccentricities. By systematically labeling these datasets, researchers are able to build predictive and diagnostic models that can be benchmarked and validated across open platforms [34].

A variety of sensors have been employed for capturing these fault-related signals. Among them, vibration sensors are most commonly used due to their sensitivity in identifying mechanical degradation. Micro-Electro-Mechanical System (MEMS) accelerometers, in particular, have emerged as a suitable option because of their small size, low cost, wide frequency response, and low power consumption, making them favorable for both laboratory and industrial applications [35,36]. Moreover, MEMS accelerometers can be easily integrated into Internet of Things (IoT)-based predictive maintenance frameworks, thereby enabling scalable and real-time condition monitoring [37].

The AC Machinery Fault Simulator is one such widely used platform, which allows for the controlled introduction of both electrical and mechanical faults, thereby facilitating the rapid collection of reliable and repeatable datasets [38]. High-resolution data acquisition systems with precise synchronization across sensors further enhance fault signature quality, providing rich datasets for developing advanced ML and deep learning models [39].

To ensure reproducibility and to accelerate research, several large-scale publicly available projects and datasets have been established:Case Western Reserve University (CWRU) Bearing Dataset: One of the most widely cited repositories, containing vibration signals acquired from a 2-hp Reliance electric motor with artificially induced bearing faults under different loads and operating conditions. The dataset is freely available online, and has become a de facto benchmark in condition monitoring research [40].Paderborn University Bearing Dataset: A comprehensive dataset containing run-to-failure experiments of bearings under real operating conditions, capturing vibration, current, and speed signals. It also provides high-resolution failure progression data, making it suitable for prognostics studies. The dataset can be accessed at the official Paderborn repository [41].NASA Ames Prognostics Data Repository (IMS Dataset): Managed by the NASA Prognostics Center, this repository includes multiple bearing run-to-failure datasets (the IMS dataset) acquired under controlled laboratory conditions with seeded defects. Signals include vibration and temperature, and the dataset has been extensively used for developing prognostics and health management (PHM) models [42].MAFAULDA Dataset (Machinery Fault Database): A large and diverse data collection initiative from the Polytechnic University of Madrid containing vibration, acoustic, and motor current signals, covering a wide range of IM conditions and fault scenarios [43].

### 3.2. Data Collection Through Modeling for Fault Diagnosis in Induction Motors

Using mathematical tools to portray a system is an approach known as modeling. The system can be simulated thanks to this representation, which offers important insights into how it functions and behaves. There are several benefits to modeling, especially when developing complicated systems [44]. Modeling improves knowledge, maximizes performance, and minimizes risks while drastically cutting development time and costs by offering a virtual environment for testing and analysis. It will remain essential as technology develops for the creation and use of creative and effective solutions in a variety of sectors. Simulation has emerged as a powerful tool to analyze machine behavior under different fault conditions, enabling in-depth study without the need for physical experimentation [45].

#### 3.2.1. Modeling Air Gap Variations for Fault Analysis in IMs

The symmetrical nature of the IMs offers a significant advantage for fault detection. Any deviation in this feature changes the interaction between the stator and rotor fluxes, causing disturbances in the stator current and the motor vibration. One promising modeling approach to IM fault detection is through analysis of the motor’s air gap, as changes in the air gap can indicate the presence of IM faults [45]. Any fault in the motor can cause changes in the motor’s airgap characteristics, particularly in the air gap permeance and the Magnetomotive Force (MMF). As shown in Figure 3, the process is explained step by step. These parameters affect the stator’s mutual and self-inductance, influencing the stator current. This modeling approach is based on geometry and relies on detecting eccentricity. Therefore, the goal here is to establish a distribution in the motor gap and gather stator current data. Extensive research has explored methods for modeling air gap variations and relating them to bearing health [46]. Broken rotor bars, which can lead to air gap irregularities, have been a particular focus of induction motor fault analysis. Algorithms suitable for real-time implementation have been developed to detect changes in parameters like mass unbalance, stiffness, and damping that may indicate bearing issues. Similarly, studies have shown that mechanical faults like bearing failures can influence the stator current, magnetic field distribution, and other measurable quantities, providing avenues for fault detection [47,48]. By modeling the relationship between air gap variations and IM health, researchers have made progress toward reliable, noninvasive CM of IMs.

#### 3.2.2. Electromagnetic Model for Fault Detection in IMs

To address this issue, researchers have explored using electromagnetic signatures, such as stator current, vibration, and stray flux, to detect and diagnose faults in induction motors. An approach, known as model-based fault detection, involves developing algorithms that can identify specific fault characteristics in the electromagnetic signatures of the machine. These algorithms can be customized to detect specific types of fault, such as raceway faults in rolling element bearings, changes in mass imbalance, and variations in stiffness and damping. As shown in Figure 4 Finite element analysis (FEA) of a faulty induction motor can also provide valuable information on the effects of faults on the machine’s electromagnetic field, which cannot be observed directly [49].

FEAs, coupled with the partial element equivalent circuit (PEEC), offer several advantages and perfectly emulate the behaviors of the equipment prototype [46]. The FEM is an effective method for computing local values in electromagnetic analysis. These local values represent the flux intensity and current density. By combining this method with circuit analysis using PEEC, we can obtain global values such as voltage and current [51].

### 3.3. Comparative Analysis of Real and Simulated Data Collection

A meaningful comparison between real data collection and modeling-based data generation for induction motor (IM) fault diagnosis requires the consideration of several criteria. In terms of accuracy and realism, real data offers the advantage of capturing the true motor behavior, including noise, nonlinearities, and unexpected interactions. This makes it highly realistic, though often accompanied by uncontrollable disturbances. Conversely, modeling produces clean and controlled signals under idealized assumptions. While this allows specific fault scenarios to be simulated with precision, it may not fully represent the complexities of real-world operating environments. Cost and practicality also distinguish the two approaches. Real data collection typically demands costly test rigs, multiple motors, sensors, data acquisition systems, and, in some cases, destructive testing (e.g., intentionally damaging rotor bars or bearings). By contrast, modeling is more economical and flexible, requiring no physical hardware and enabling safe simulation of a wide range of faults. Reproducibility presents another point of divergence. Real data is difficult to replicate exactly because of variations in environment, motor aging, and load fluctuations (such as overlapping torque and fault frequencies). Simulated data, on the other hand, is fully reproducible since it is generated from predefined models and parameters. Fault coverage and dataset scalability are equally important considerations. Real data is inherently limited by practical constraints, as not all faults, especially severe ones like winding breakdowns, can be tested safely. Modeling, however, makes it possible to simulate any type of fault, from mild to catastrophic. Moreover, collecting large datasets under varied operating conditions is time-consuming and resource-intensive in experimental setups, whereas modeling enables the generation of extensive and balanced datasets with controlled variations in load, speed, and fault severity. Perhaps the most critical criterion in the context of this review is the applicability to ML and deep learning. Real data is more reliable for training models that generalize well to practical systems, even though available datasets are often small. Simulated data is particularly useful for pre-training and validation, but it may introduce a “reality gap”, necessitating transfer learning or domain adaptation when applied to actual motors. Overall, modeling appears to surpass real data collection in many of the considered criteria. Nevertheless, the two approaches should be viewed as complementary rather than competing. Future research should focus on generating more model-based data informed and calibrated by existing benchmark datasets, thereby constructing meta-data frameworks capable of tuning the governing equations and correlations of IM physical phenomena with greater accuracy.

### 3.4. Data Acquisition Modalities in Induction Motor Monitoring

Conventional fault detection techniques for induction motors rely on various monitoring approaches, each targeting specific fault signatures [52]. Vibration analysis is widely used to detect mechanical anomalies such as bearing defects, rotor imbalance, or misalignment [53]. Current signature analysis examines the stator current to identify electrical and electro-mechanical faults, including broken rotor bars or stator winding issues. Temperature monitoring provides insights into thermal stress and overheating, which may indicate insulation degradation or bearing wear. Acoustic emission monitoring captures high-frequency sound waves generated by faults, offering early detection of mechanical defects. Additionally, rotor speed and torque observation can reveal irregularities in motor performance caused by mechanical or electrical faults. Although vibration-based sensing is widely adopted, complementary data acquisition strategies are also utilized. Current signature analysis, for instance, provides a non-intrusive and cost-effective alternative for monitoring electrical faults such as stator winding inter-turn short circuits and broken rotor bars [54]. Emerging modalities, such as thermal monitoring, acoustic emission, and electromagnetic flux analysis, are increasingly being explored to complement traditional methods and enhance fault diagnosis capabilities. Nevertheless, motor current signature analysis (MCSA) and vibration analysis (VA) remain the leading approaches for reliable and efficient condition monitoring, with MCSA providing low-cost detection of electrical faults through stator current analysis [55], and VA offering highly sensitive detection of mechanical faults based on extensive practical experience [56].

### Literature Search Methodology

To collect relevant academic references, two methods were used:1.Automated Scraping Attempt.

A Python 3.10 script was developed to scrape academic databases (IEEE Xplore, Piscataway, NJ, USA, Elsevier, Amsterdam, the Netherlands) for articles matching specific keywords. However, due to technical challenges such as access restrictions and CAPTCHA verification, this approach was unsuccessful [57].

2.Manual Literature Search.

The search was conducted manually by entering keywords such as “Current analysis” and “Vibration analysis” into the search engines of IEEE Xplore and Elsevier to compare the results shown in Table 3. Articles were filtered by publication year (2013–2023), and the number of occurrences for each keyword was recorded.

MCSA has a relatively smaller number of publications in both IEEE and Elsevier, suggesting that it may be less commonly used for fault detection in induction motors. MCSA is a more recent technique that has gained popularity in recent years, as it can be used to detect electrical faults such as stator winding faults and broken rotor bars. Current analysis is a relatively new technique, and the number of publications in this area will likely continue to grow in the future. The survey results indicate that vibration analysis is the most widely used method for fault detection in induction motors. Researchers and practitioners in this field may benefit from considering vibration analysis as a primary tool for fault detection and diagnosis.

### 3.5. Multimodal Data Acquisition for Induction Motor Fault Diagnosis

Multimodal data acquisition refers to the collection of diagnostic information from multiple heterogeneous sensors, capturing different aspects of induction motor operation such as electrical signals (current, voltage), mechanical responses (vibration), thermal characteristics (temperature), and acoustic emissions. By integrating these complementary modalities, multimodal acquisition provides a richer representation of the motor’s health state than single-sensor approaches [58]. Modern industrial systems commonly deploy such sensors under the Industrial Internet of Things (IIoT) paradigm to ensure synchronous, high-fidelity data collection for use in ML and physics-informed diagnostic methods [59].

The key advantage of multimodal data acquisition lies in its ability to capture complementary and redundant fault information across diverse physical domains, which improves robustness and diagnostic accuracy. For instance, vibration signals are sensitive to mechanical misalignment or bearing faults, while stator current analysis better reflects electrical faults. By fusing these modalities, a more holistic fault profile is achieved [60]. In addition, multimodal data helps mitigate the limitation of any individual sensor, reducing the risk of false alarms or missed detections under complex and variable operating conditions [61]. Studies have also shown that multimodal fusion can improve early fault detection, particularly in incipient or compound fault scenarios where fault signatures may be weak or distributed across domains [62].

Despite its benefits, multimodal acquisition introduces challenges in terms of system complexity, cost, and data management. Deploying multiple high-frequency sensors increases hardware and maintenance costs and can raise synchronization issues, as signals collected at different sampling rates must be aligned carefully for meaningful analysis [63]. In addition, multimodal data streams are often high-dimensional, leading to computational burdens and potential overfitting if not managed with effective feature selection or dimensionality reduction. Interoperability across heterogeneous sensor types and legacy systems adds another layer of complexity in industrial deployments.

To address these challenges, recent works have proposed multimodal data fusion frameworks that combine sensor inputs at feature, decision, or model levels to streamline integration [64]. Dimensionality reduction and representation learning techniques, such as autoencoders or attention-based networks, are used to capture shared latent features that reduce computational cost while retaining salient fault information [65]. When full multimodal deployments are not feasible, virtual sensing and physics-informed models can provide surrogate measurements, enhancing monitoring without costly sensor installations. These strategies allow multimodal fault diagnosis to remain both practical and effective in real-world industrial IM monitoring.

## 4. Fault Signature Extraction and Signal Processing

Fault signature extraction is a crucial step in IMs’ CM. It aims to transform raw measurements into discriminative representations that facilitate the identification and classification of faults. Advances in computer science and applied mathematics have made signal processing techniques increasingly sophisticated and effective. When applied to CM, SP serves as a powerful tool for feature extraction, helping to reduce both computational resources and the time required for the final fault detection stage, ultimately enhancing the accuracy and performance of CM systems. In modern CM architectures, SP is often considered a preprocessing step. However, with the emergence of deep learning (DL) methods, there is a growing trend toward minimizing or even eliminating this step, as DL models can learn to extract features directly from raw data. This section reviews the major approaches used for feature extraction, ranging from traditional signal processing techniques to modern ML-based strategies. In the following section, we present the various methods employed in the core part of the CM process. A brief overview of each approach is provided, followed by a discussion of the challenges associated with data-driven techniques: High-dimensional data is a common feature in modern ML applications and present both opportunities and challenges. Data scientists can unlock the value of these complex datasets using the right tools and techniques, such as dimensionality reduction and feature selection [66]. As technology develops, effective working with high-dimensional data will become essential to driving innovation and discovering new insights in different fields. Imbalanced Data is a common problem in ML where the training data is not evenly distributed across all classes or categories. This can result in biased models that do not perform well in the minority class and may not generalize well to real-world data [67]. Nonlinearity refers to the property of a system or relationship that does not exhibit a linear or directly proportional relationship between the input and output. In other words, the output is not only a direct multiplier of the input [68]. Noise and overfitting are two related concepts in the context of empirical data analysis, particularly in ML and statistical modeling [69].

### 4.1. Time-Domain Feature Extraction

Time-domain features are directly derived from the raw signals and remain the most intuitive descriptors of motor behavior. Commonly used features include the mean, root mean square (RMS), standard deviation, peak-to-peak amplitude, skewness, and kurtosis. These values are obtained by dividing the complete vibration signal into one-second segments. For each segment, the corresponding features are computed across the entire eighteen-second vibration recording. Subsequently, all extracted features are concatenated to form a single dataset representing the time-domain characteristics of the raw vibration signal [70]. These features are simple to compute and provide useful statistical information, but they may fail to capture hidden periodicities or frequency-related signatures in the data.

### 4.2. Time Frequency Feature Extraction

Frequency-domain techniques rely on spectral analysis to reveal periodic components that are often associated with specific fault mechanisms. Fourier Transform (FT) and its variants are widely applied to decompose signals into their constituent frequencies. Fault signatures such as sideband harmonics, frequency peaks, or energy distribution across spectral bands can be effectively identified in this domain. Although powerful, FT-based methods assume stationarity and may be less effective for non-stationary fault signals. To handle non-stationary signals, time–frequency approaches such as Short-Time Fourier Transform (STFT), Wavelet Transform (WT), and Hilbert–Huang Transform (HHT) are employed. These methods allow simultaneous localization in both time and frequency, making them well suited for transient fault detection. Wavelet-based features (e.g., wavelet energy, entropy, and coefficients) are particularly effective in isolating impulsive patterns caused by localized defects.

#### 4.2.1. Fast Fourier Transform

The FFT, in its various forms (Classical, Instantaneous Power FFT, Bispectrum, etc.), has been widely utilized in fault signatures stigmatization for VA and MCSA. FFT has limitations, such as masking characteristic frequencies by the supply frequency and the inappropriateness of transient signals [71]. To overcome this kind of problem, newer techniques used for signal processing include a Short-Time Fourier Transform (STFT), Wavelet Transform (WT), wavelet packet decomposition (WPD), Wigner Ville distribution (WVD), power spectral density (PSD), support vector machines (SVMs), Prony analysis, fractal analysis, and fuzzy logic [71]. Despite their effectiveness, these methods are primarily suited for non-stationary signals. Recent research has therefore focused on utilizing both starting and steady-state current signatures to detect faults, including broken rotor bars. In addition, finite element analysis and Wavelet Transform have been used to characterize internal motor faults, offering a more detailed insight into motor behavior under faulty conditions [72]. Recent advances in signal processing have led to the exploration of novel methodologies, including the Park transform and artificial neural networks, for the analysis of stator current signatures, thus offering alternative approaches to traditional techniques [73].

#### 4.2.2. Wavelet Transform (WT)

For an accurate VA and MCSA signal analysis, the Wavelet Transform was introduced with the idea of overcoming difficulties encountered in the application of FFT [74]. The WT is a windowing technique with a variable-size region is then used to perform the signal analysis, which can be the stator current. By using wavelet analysis, we can obtain more precise low-frequency information by using long-time intervals, and more accurate high-frequency information by using shorter regions. The ability to perform local analysis is one of the most interesting features of the Wavelet Transform [75].

#### 4.2.3. Hilbert Transform (HT)

The Hilbert Transform (HT) is a widely used and influential method in signal analysis, with applications spanning various scientific domains, including fault diagnosis, signal transmission, geophysical data processing, and the detection of mechanical load anomalies and rotor cage faults in induction motors [76]. The HT complements other signal processing techniques, such as the Fast Fourier Transform (FFT) and Wavelet Transform (WT), by providing enhanced capabilities for analyzing non-stationary signals and extracting instantaneous frequency and phase information [77]. This approach enables the extraction of meaningful features from raw data, reducing its dimensionality and facilitating more accurate analysis.

As demonstrated in [78], the HT can be improved through the use of Estimation of Signal Parameters via Rotational Invariance Techniques (ESPRIT) for detecting rotor faults in induction motors at low slip. Similarly, the author in [77] showed that the HT can be used to enhance the resolution of the MCSA method, enabling the diagnosis of rotor asymmetries at very low slip. These studies highlight the potential of the HT in fault diagnosis and demonstrate its effectiveness in various applications.

The HT, FFT, and WT are all computationally intensive methods that require the solution of complex mathematical equations, which can be a significant limitation. However, the integration of ML Techniques can mitigate this limitation, offering several benefits, including the ability to handle large volumes of data, expedited processing, and early detection of faults [78]. Using ML techniques, researchers, and practitioners can overcome the computational challenges associated with traditional signal processing methods and develop more efficient and effective FD systems.

While time–frequency methods provide rich representations of non-stationary signals, they often suffer from high dimensionality, sensitivity to noise, and parameter dependence. Statistical feature extraction has emerged to address these limitations, offering compact, interpretable, and computationally efficient descriptors that improve fault classification.

### 4.3. Statistical Feature Extraction

Statistical descriptors are often employed to summarize signal distributions in either the time or frequency domain. Higher-order statistics (e.g., skewness, kurtosis, entropy) and correlation-based measures provide additional discriminatory power. Furthermore, advanced statistical learning methods such as principal component analysis (PCA) or independent component analysis (ICA) can be applied to reduce dimensionality while preserving fault-relevant information.

In the diagnosis of IM, vibration signals are commonly monitored as a key parameter, with the root mean square (RMS) value of vibration velocity, crest factor, and kurtosis being the primary indicators used to detect anomalies [79].

The root mean square (RMS) value reflects the energy content of the vibration signal and is defined as(1)$$\mathrm{RMS}=\sqrt{\frac{1}{N}\sum_{i=1}^{N}x_{i}^{2}}$$
where $x_{i}$ is the vibration amplitude at sample *i*, and *N* is the total number of samples.

The crest factor (CF) provides sensitivity to impulsive events and is expressed as(2)$$\mathrm{CF}=\frac{\max|x_{i}|}{\mathrm{RMS}}$$

A high CF indicates the presence of sharp transients, often linked to bearing defects or mechanical impacts.

The kurtosis (K) is a fourth-order statistical moment measuring the peakedness of the signal distribution:(3)$$K=\frac{\frac{1}{N}\sum_{i=1}^{N}{\left(x_{i}-\mu \right)}^{4}}{{\left(\frac{1}{N}\sum_{i=1}^{N}{\left(x_{i}-\mu \right)}^{2}\right)}^{2}}$$
where μ is the mean of the signal.

In the CM context the rms value is a measure of the energetic dissipation caused by dissipative events, providing insight into the overall energy content of the vibration signal. According to [80,81], the recommended rms thresholds have been established to facilitate the detection of faults. The crest factor, calculated as the ratio of the peak amplitude of the vibration signal to its normalized RMS value, is a sensitive indicator of impulsive events, such as those resulting from collisions between bearing components [82,83]. This measure allows for the assessment of the significance of these events relative to the overall RMS value of the signal, enabling the identification of potential faults [84]. As noted in [85], the crest factor is particularly useful in detecting bearing failures, a common cause of IM failures. Furthermore, [86] demonstrated that the combination of RMS and crest factor analysis can improve the precision of fault diagnosis in IM.

Statistical feature extraction emerged as a means to overcome some of the limitations of traditional time–frequency techniques, particularly their high dimensionality, sensitivity to noise, and parameter dependence. By condensing raw signals into compact descriptors such as mean, variance, RMS, skewness, and kurtosis, statistical approaches offer both computational efficiency and physical interpretability. However, these handcrafted features still rely heavily on expert knowledge and may fail to capture complex, nonlinear dependencies inherent in real-world induction motor data. To address these shortcomings, ML and, more recently, deep learning-based feature extraction methods have been developed. Classical MLAs, such as support vector machines (SVMs) and Random Forests (RFs), can automatically identify discriminative combinations of handcrafted features. In contrast, deep learning models including convolutional neural networks (CNNs), recurrent neural networks (RNNs), and Transformers can learn hierarchical and task-specific features directly from raw signals, thereby minimizing the need for manual feature engineering and offering superior generalization across diverse operating conditions.

### 4.4. Machine Learning and Deep Learning-Based Feature Extraction

With the advent of data-driven methods, the extraction of features in IM diagnosis has increasingly shifted toward automated learning. ML algorithms such as SVM, RF, and k-Nearest Neighbor (kNN) can operate on handcrafted features while also assisting in feature selection. More recently, deep learning approaches such as CNN, RNN, and Transformers have demonstrated the ability to learn hierarchical feature representations directly from raw signals, reducing reliance on manual feature engineering and often yielding superior performance in complex fault scenarios.

Several studies have adopted a two-stage approach, where one ML or DL model is used for feature extraction and another for classification. Convolutional Autoencoders (CAs) have been used to extract features from current and vibration signals, followed by SVM for classifying stator turn faults and broken rotor bars in IMs, achieving over 95% in [87]. In [88], Triplet CNN embeddings combined with SVM have successfully addressed rolling bearing fault classification under small-sample conditions. In [89] Unsupervised Autoencoders (UAs) have been employed to select features from vibration data, followed by feed-forward neural networks for motor bearing fault classification. The author in [90] presented a nonlinear feature extraction using (Kernel) PCA or ICA followed by SVM has been applied for multiple types of IM faults, representing a classical two-stage pipeline. Other approaches include component-analysis-based features on transient signals classified with SVM [91], wavelet packet transform combined with PCA feeding a dual SVM for IM bearing faults [92], and CNN autoencoders for feature extraction followed by deep neural networks for multimodal electric motor fault diagnosis, integrating both signal and power analysis data [93].

### 4.5. The Efficiency Signal Processing Approach

The CM process involves two critical steps, each requiring careful selection of the most suitable approach. These steps are signal processing and fault detection using intelligent systems. SP and feature extraction play a key role in reducing noise and dimensionality by extracting relevant features from the data. Subsequently, ML and DLM techniques are employed to identify patterns and correlations that indicate potential faults.

The FFT converts signals to the frequency domain, but often results in a high number of frequency components, which requires feature selection or dimensionality reduction techniques like PCA [94]. However, FFT does not inherently handle class imbalance, requiring post-processing methods such as resampling or weighting [95]. Additionally, FFT assumes stationary and linear signals, making it less effective for highly nonlinear faults, and it is sensitive to noise, often requiring prefiltering (e.g., low-pass filters) to mitigate noise impact [96]. In contrast, the WT produces a large number of coefficients, but these can be managed through feature selection techniques like statistical moments of wavelet coefficients [97]. Although WT does not inherently address class imbalance, extracted features can be processed with balancing techniques [98]. WT outperforms FFT in handling nonlinear and non-stationary signals and reduces noise sensitivity through the use of different mother wavelets (e.g., Daubechies, Morlet) [99]. HT generates additional features, which can be managed with proper feature selection, but it requires external techniques like Synthetic Minority Over-sampling Technique (SMOTE) or cost-sensitive learning to handle class imbalance [100,101]. HT works well with nonlinear signals when combined with methods like Empirical Mode Decomposition (EMD) [102]. Although it is sensitive to noise and is often used with band-pass filtering to enhance fault signatures [103]. WT is a powerful tool for analyzing transient signals; however, the selection of the mother wavelet is arbitrary and can introduce errors. For this purpose, this approach is primarily used as a complementary tool to other techniques, mainly ML-based methods [104]. In summary, time-domain, frequency-domain, and time–frequency techniques have long served as the foundation for fault feature extraction, each providing complementary insights into induction motor behavior. However, their effectiveness is often constrained by sensitivity to noise, limited capacity to capture nonlinear dynamics, and challenges in handling high-dimensional data. To overcome these issues, statistical feature extraction has been developed, offering more compact and discriminative representations. Building on this foundation, ML approaches further enhance feature extraction, while deep learning models advance the field by enabling end-to-end diagnosis directly from raw signals. Together, these methodologies represent an evolutionary trajectory that balances interpretability, robustness, and accuracy in condition monitoring systems.

## 5. Machine Learning and Deep Learning-Based IMs Fault Classification

Classification is a fundamental and widely understood function in ML. It requires making predictions about the category or class of certain data items using labeled data for training. Nevertheless, classification as the “most basic” function can be deceptive given the intricacy that can manifest itself in real-world scenarios. The classification is conceptual simplicity at its core, it is about assigning labels to inputs, for example, determining whether an email is spam or not, recognizing handwritten digits, or in our case, fault detection in IM [76]. To efficiently tackle fault classification, researchers have devised a range of classification algorithms that can be classified into several categories, such as Linear Models, which encompass logistic regression and linear discriminant analysis techniques. Random Forests and gradient-boosting machines are two such instances of tree-based algorithms in decision tree analysis. [86] highlights the possibilities of applying the RF in machine fault diagnosis, proposing a hybrid method combined with a genetic algorithm to improve classification accuracy. There are also support vector machines (SVMs), which are highly efficient in handling high-dimensional spaces. In [105], a diagnosis was made on the data collected using SVM, multilayer neural network, convolutional neural network, gradient boosting machine, and XGBoost ML models. An inherent limitation of employing SVM is that it performs effectively when there is a distinct boundary between two classes. In such cases, the algorithm must address the quadratic optimization problem to identify the ideal hyperplane separating the two classes [106]. Another kind of algorithm is the k-Nearest Neighbors (KNN), which is a straightforward model for instance-based learning. Utilizing KNN is inefficient due to its being the least robust algorithm, highly susceptible to outliers, and its accuracy is contingent upon the selection of k [107]. More sophisticated ML techniques are neural networks encompassing deep learning models designed for intricate tasks like acoustic and image recognition. In [108], various ANN models, including multi-layer perceptron (MLP) and Radial Basis Function (RBF), have been investigated for bearing fault detection in induction motors, including single-phase monitoring. However, choosing the “best” one depends on various factors. The literature study indicates that deep learning models, especially CNNs [109], regularly surpass standard MLAs such as MLPs and networks in IM defect identification. However, owing to the substantial computing demands of CNNs, we intend to identify a model that accomplishes efficient fault detection while utilizing fewer resources. In the following section, we present a curated selection of ML algorithms that can be effectively employed to develop an intelligent CM system. Our objective is to highlight the most commonly used algorithms in the literature, with a particular focus on two major categories: ANN-based methods and decision tree-based ensemble techniques.

### 5.1. Radial Basis Function (RBF) Networks

Radial Basis Function (RBF) networks are a distinct category of artificial neural networks known for their exceptional precision in modeling intricate nonlinear interactions. Thus, they are particularly adept for applications that require the identification and analysis of faults [110,111]. A primary advantage of employing RBF networks for IM failure detection is their ability to understand complex relationships between input parameters, including vibration, stator current, stray flux, and the associated bearing fault conditions. In [112], different impacts of neural network complexity and RBF activation function characteristics on data classification quality were illustrated. The RBF Kernel can be used in combination with Polynomial to minimize error classification such as in [113]. Recent studies indicate that Radial Basis Function networks could be effective in identifying bearing defects in induction motors.The architecture of the RBF network is illustrated in Figure 5.

The use of RBF networks to identify defects in induction motors has shown considerable promise. These networks may effectively represent the complex relationships between input data and fault circumstances, facilitating accurate and reliable problem identification, especially in challenging operational environments. To summarize, the primary benefits of employing RBF networks for detecting IMs faults encompass their capability to represent intricate nonlinear correlations between input variables and fault situations [114]. This model allows for elevated precision in identifying diverse IM types, including instances of multiple faults. Furthermore, RBF shows a resilience to noise and other operating challenges. The effectiveness of this method was confirmed through practical trials carried out under various bearing failure conditions [115]. Nevertheless, there are several constraints to using RBF networks in IM defect detection, including a substantial demand on training data to reach the best performance. The model may be computationally demanding, particularly for extensive applications. The RBF needs a special learning low to adjust the node number in the hidden layer automatically as well as a small number of epochs. These advantages are not enough to consider them more powerful than other classical ANNs such as the MLP structure [115]. This could lead to a substantial demand for meticulous design and selection of features to ensure the use of the most relevant inputs for the network [116,117].

### 5.2. Multi-Layer Perceptron (MLP)

The MLP has an input layer, one or more hidden layers, and an output layer. Throughout the training process, the network acquires the ability to associate the input characteristics with their respective fault classes through backpropagation of the error gradient [118]. The MLP is a promising deep learning methodology, a variant of artificial neural networks capable of learning intricate nonlinear correlations from data [119]. It is trained on a labeled dataset, whereby each data point comprises an input vector and its associated output vector. The MLP acquires the ability to correlate input vectors with output vectors by modifying its internal weights and biases. It can effectively handle high-dimensional input data, such as vibration or current signals, and extract relevant features for fault classification [120]. The application of MLP for bearing fault detection in induction motors has been carefully examined in the literature. Sadeghi in [121] proposed a multistream convolutional neural network that integrated data from motor vibration and stator current signals, exhibiting enhanced performance compared to traditional ML techniques. More in depth, Grezmak in [122] presented an exhaustive overview of the use of deep learning techniques, including multi-layer perceptrons, for the detection of electrical motor defects, highlighting the current state of the art and the prospects for further developments in this domain. In bearing defect detection, the multi-layer perceptron can extract pertinent information from motor vibration or current signals and subsequently categorize the fault. In contrast to conventional ML methods, an MLP can autonomously acquire the best features for fault classification, eliminating the need for manual feature engineering. Figure 6 presents a schematic illustration of a simple MLP network.

Researchers have investigated several ways to improve the classification performance of MLP [123,124]. That includes integrating specialized network topologies that utilize the spatial and temporal correlations present in the input data.

### 5.3. Decision Trees (DTs)

A decision tree is a traditional ML model that is derived inductively from a set of samples. In the tree, each node represents an attribute (or characteristic), while the edges denote values (or ranges of values) linked to that property as illustrated in Figure 7 [125]. The appearance of an attribute in a tree indicates the relevance of the associated attribute [126,127]. Several DT algorithms have been developed, including Conditional Inference Trees [100,128], Chi-squared Automatic Interaction Detection (CHAID) [129], C4.5 [130], as well as classification and regression trees (CARTs) [131].

### 5.4. Random Forest

In the field of IM failure identification, the Random Forest Algorithm (RFA) is a powerful technique that has attracted a lot of interest. To detect and diagnose mechanical problems in rotating equipment, such as increased mass unbalance, raceway faults in rolling element bearings, and variations in stiffness and damping, this program employs a model-based approach [86]. The technique has been thoroughly examined and evaluated using computer simulations, demonstrating that it is a successful real-time implementation; see Figure 8 for the Random Forest workflow. Numerous industrial applications, especially those that involve weapons and equipment that must withstand high temperatures, high speeds, and large loads, frequently experience bearing problems. Researchers have developed several data-driven strategies, such as ML techniques, to solve this problem [2].

As shown in [132,133], RF has been used to address various IM failure conditions. Furthermore, a thorough analysis of the state of the art in deep learning-based IM fault diagnostics is provided in the survey work by [134], which also highlights the advantages and disadvantages of various algorithms.

### 5.5. Emerging Trends and Future Possibilities

Over the past decades, IMs fault diagnosis has evolved from traditional signal processing methods toward sophisticated data-driven and hybrid techniques. With increasing industrial demands for reliability, energy efficiency, and predictive maintenance, several emerging trends are shaping the future of IM condition monitoring and diagnostics:

#### 5.5.1. Convolutional Neural Networks (CNNs)

CNNs represent a class of deep learning models that are particularly effective in image recognition and spatial data analysis. The foundational concept can be traced back to the work of Fukushima [135], who proposed an early hierarchical model capable of recognition of shift-invariant patterns. This idea was extended and applied effectively by LeCun [136], who introduced the LeNet architecture and demonstrated its success in handwritten digit recognition tasks. The popularity of CNNs increased significantly with the introduction of AlexNet by [137], which achieved outstanding performance in the ImageNet Large Scale Visual Recognition Challenge (ILSVRC). This breakthrough marked the beginning of wide adoption of CNNs for computer vision tasks. Subsequent architectures, such as VGGNet [138], further extended network depth and influenced many modern CNN designs.

Deep learning-based methods have shown promising results in the diagnosis of IM faults, as they can effectively extract features from raw sensor data without requiring extensive feature engineering [2]. CNNs have shown their ability to extract relevant features from vibration, current, or acoustic emissions data, making them an appropriate choice for IM fault detection [139]. These deep learning models (DLMs) can learn classified representations of input data, catching both local and global patterns that are indicative of fault detection [140]. Unlike traditional approaches that rely on handcrafted feature extraction and require substantial expertise in data analysis and signal processing. CNNs can directly process time-series or two-dimensional signal representations such as spectrograms, thereby reducing reliance on domain-specific signal processing [141]. One major advantage of CNN-based diagnosis is their high classification accuracy, even under noisy or variable operating conditions. Several studies have demonstrated that CNNs outperform traditional ML methods when dealing with complex, nonlinear patterns in motor fault data [140]. CNNs are also scalable, enabling the same architecture to be applied across different machine sizes and fault categories with minimal redesign, which underscores their reliability and practicality for industrial applications [142,143]. Furthermore, CNN architectures allow for real-time deployment due to their efficient inference capabilities when implemented on modern GPUs or edge devices. This computational efficiency, combined with their robustness and precision, makes CNNs particularly suitable for PM frameworks where early detection of motor faults is critical for minimizing downtime and operational costs [144].

To illustrate this process, in Figure 9, we have a schematic representation of a CNN architecture. The network begins with an input image, followed by multiple convolutional and pooling layers for feature extraction. The feature maps are then flattened and passed through fully connected layers with dropout regularization to prevent overfitting. The final output layer uses a softmax activation function for classification.

However, these conventional approaches often struggle to extract meaningful features from raw data effectively [145,146]. To address these limitations, other DLMs have emerged as a powerful alternative, offering superior feature extraction capabilities compared to traditional ML algorithms [147].
Figure 9A schematic representation of a CNN architecture [148].
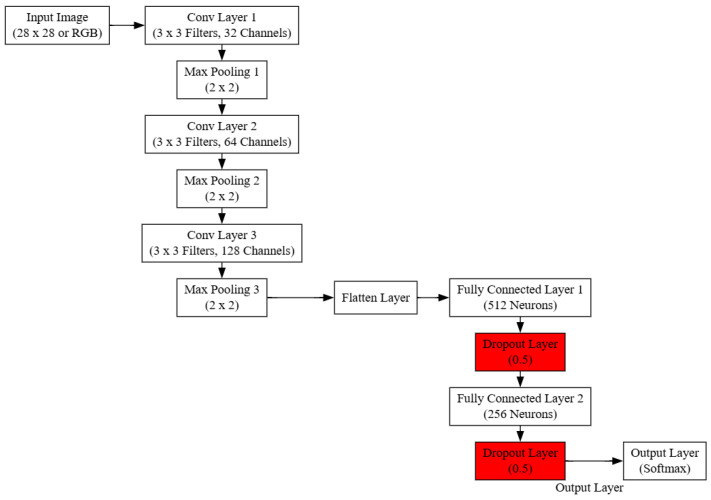


#### 5.5.2. Recurrent Neural Networks (RNNs)

RNNs represent a class of DLM architectures capable of handling sequential data of arbitrary length and have been effectively applied to a wide range of end-to-end learning tasks. Time-series modeling using RNNs and their variants has attracted increasing interest, as the sequential and temporal characteristics of motor signals can be effectively used for condition monitoring and fault diagnosis [149]. Standard RNNs suffer from vanishing and exploding gradient problems, which limit their ability to capture long-term dependencies. To address this limitation, Hochreiter and Schmidhuber introduced the Long Short-Term Memory (LSTM) network, which uses gated mechanisms (input, forget, and output gates) to regulate information flow and preserve relevant context over longer sequences [150]. LSTM have proven advantageous in handling motor signals due to their ability to mitigate vanishing gradient issues and capture both short- and long-term dependencies in sequential data [151]. This makes it particularly suitable for analyzing the complex, non-stationary time-series patterns of vibration, current, and acoustic signals generated by induction motors. Their architecture includes recurrent connections, allowing information from earlier time steps to influence later outputs.

In the context of IM fault diagnosis, LSTMs exhibit strong capabilities for modeling temporal dependencies in sensor signals, which often carry fault information over extended time horizons. By leveraging their gating mechanisms, LSTMs effectively retain fault signatures while filtering out noise and irrelevant fluctuations [152]. Compared to traditional ML and statistical methods, LSTMs require less manual feature engineering, as they directly learn representations from raw or minimally preprocessed signals [153]. They also demonstrate robustness under varying load and speed conditions, enhancing generalization to different operational scenarios [154]. Empirical studies show that LSTMs outperform conventional classifiers and even shallow neural networks for incipient fault detection, making them a strong candidate for PM [155].

Despite their advantages, LSTMs face certain limitations in industrial fault diagnosis. First, they are computationally more expensive than classical models, especially for long sequences. Training LSTMs requires careful hyperparameter tuning (e.g., number of units, dropout rates) to avoid overfitting when fault datasets are small. Additionally, while they capture temporal relationships, they may fail to exploit global contextual information as efficiently as attention-based models such as Transformers. Another challenge is their comparative “black-box” nature, which limits interpretability and may hinder industrial acceptance in safety-critical applications.

Several strategies have been proposed to mitigate these issues. Data augmentation and transfer learning improve LSTM performance in scenarios with limited training samples [146]. Hybrid approaches, where LSTMs are combined with CNNs, are increasingly adopted for IM fault diagnosis, as CNNs capture local spatial features while LSTMs model temporal dependencies [149]. More recently, attention-augmented LSTM architectures have been investigated to enhance global feature capture and interpretability, bridging part of the performance gap with Transformer models. Thus, while some limitations remain, LSTMs continue to offer a balanced and practical approach to fault diagnosis in induction motors.

#### 5.5.3. Transformer Architectures

Transformer architectures, originally introduced for natural language processing by [156], rely on self-attention mechanisms that capture long-range dependencies in sequential data without the recurrence operations used in traditional RNNs. Their ability to model contextual relationships efficiently has led to rapid adoption in computer vision, signal processing, and time-series analysis. Recently, variants such as Vision Transformers (ViT) [157] and Time-Series Transformers [158] have been applied to industrial monitoring tasks, including condition-based maintenance and machinery fault diagnosis. When applied to induction motors, Transformers use their attention layers to extract fault-relevant patterns from vibration, current, or acoustic signals, effectively handling both temporal and spectral representations.

Transformers offer several advantages for IM fault diagnosis compared to CNNs or RNN-based architectures. First, the self-attention mechanism enables a global view of the input, making it particularly effective for identifying subtle degradation patterns spread across long sequences [159]. Second, their flexibility allows seamless integration of multimodal inputs, such as combining current and vibration signals in a unified representation [160]. Moreover, Transformer-based models can adapt to variable operating conditions by dynamically weighting the contribution of different time points, which supports robust performance for real-world industrial scenarios. Empirical studies have shown that Transformer variants outperform or complement CNNs and LSTMs in PM settings, offering higher diagnostic accuracy and generalization capability.

Despite these advantages, some limitations to this approach have been identified. Transformers typically require large training datasets, which may not always be available in industrial settings where fault samples are scarce. They also tend to be computationally intensive, raising concerns about real-time deployment on embedded systems without hardware acceleration. Additionally, while attention mechanisms provide some interpretability, the decision-making process of Transformers is still less transparent compared to traditional physics-informed or hybrid approaches.

To mitigate these limitations, researchers have proposed lightweight Transformer variants that reduce computational cost through efficient attention approximations [161]. Transfer learning and data augmentation have also been used to address the issue of limited fault data [1,162]. In practice, a hybrid approach, combining CNNs for local feature extraction and Transformers for capturing long-range dependencies, has shown promise in balancing accuracy, interpretability, and efficiency [163]. Such strategies suggest that while Transformers alone present challenges, their integration with other methods can yield robust solutions for induction motor fault diagnosis.

#### 5.5.4. IoT, Edge Computing, and Cloud-Based Data Platforms

The proliferation of the Industrial Internet of Things (IoT) has enabled continuous monitoring of induction motors through distributed networks of sensors that collect vibration, current, temperature, and acoustic data in real time [164]. These data streams can then be transmitted for processing either at the edge, closer to the source, or in centralized cloud platforms. Edge computing refers to data processing performed locally on embedded devices or edge servers in proximity to the motor, reducing latency and bandwidth requirements [165]. Alternatively, cloud platforms provide scalable storage and computational infrastructures suited for advanced analytics and training of ML models across large datasets [166]. Together, IoT-enabled data acquisition, edge computing, and cloud integration provide the backbone of modern fault-diagnosis architectures for industrial systems.

The combined use of IoT, edge, and cloud platforms offers significant advantages in IM fault diagnosis. IoT-based sensing provides high-resolution, continuous monitoring capabilities across large fleets of motors, improving fault detection coverage. Edge computing enables real-time decision-making, allowing early response to critical events by avoiding delays due to network congestion or cloud communication [167]. Cloud computing complements this strategy by offering virtually unlimited computational resources for model training, historical trend analysis, and large-scale fleet management, thereby enhancing scalability and knowledge sharing across industrial sites [168]. This hybrid architecture balances low-latency processing at the edge with data-intensive model optimization in the cloud, ensuring both operational reliability and long-term predictive intelligence.

Despite their promise, challenges remain in deploying these technologies. IoT-based monitoring introduces cybersecurity and privacy concerns, as sensitive operational data must be transmitted across networks [169]. Edge devices, while offering reduced latency, are typically resource-constrained, limiting their ability to host complex deep learning models. Meanwhile, cloud solutions may suffer from connectivity issues and increase dependency on reliable high-bandwidth internet. Furthermore, integration of heterogeneous sensors and interoperability across platforms can be difficult in legacy industrial settings. Recent research and practice propose several strategies to overcome these barriers. Lightweight deep learning models have been adapted for deployment on edge devices, allowing efficient local inference while offloading more complex training tasks to the cloud [170]. Federated learning has emerged as a privacy-preserving alternative, enabling collaborative training across distributed IoT nodes without raw data sharing [171]. Edge–cloud collaborative frameworks, where preprocessing is done at the edge and advanced analytics at the cloud, are increasingly adopted for fault diagnosis in IMs. Ultimately, the integration of IoT, edge, and cloud platforms offers a flexible and practical infrastructure, supporting real-time monitoring while leveraging big data analytics for predictive maintenance.

## 6. Comparative Advantages of CM Approaches

Following a comprehensive review of various methods used to classify faults in IMs, this discussion section highlights the key challenges (High-dimensional, Imbalanced Data, Nonlinearity, Noise, and Overfitting) associated with fault detection and how each approach addresses them. Figure 10 is schematic representation showing how each of the reviewed methods addresses the challenges associated with data-driven IMs diagnosis, while highlighting both their respective advantages and inherent limitations.

The ML and DLM gain insights into the underlying patterns or structures that encode information within raw data. These representations, often referred to as structural descriptions or models, serve as frameworks for capturing and organizing the extracted knowledge. Once trained, such models can be used to make predictions on previously unseen data. Structural descriptions can vary in form and complexity, including representations such as decision trees where data are split based on feature-based rules and neural network weights, which define the learned parameters across layers [172]. In our review, we specifically categorized and analyzed models employing these two types of representations: decision trees (e.g., RF) and neural network weights (e.g., MLPs and CNNs), highlighting their relevance in induction motor fault detection systems. Each method offers unique strengths and limitations, making their selection dependent on the specific challenges of the fault classification task.

Table 4 presents various studies on ML- and DLM-based IM diagnosis, organized chronologically from earlier, simpler models to more recent and complex approaches. Each study outlines the complete diagnostic pipeline, revealing the diversity of methodologies employed. While earlier methods generally exhibit lower accuracy, more advanced approaches demonstrate improved performance, though they often demand greater computational resources.

Overcoming data-driven challenges in ML, especially in applications like condition monitoring (CM), requires a combination of strategies. Model like RBF networks handle high-dimensional data effectively but require careful tuning of radial basis functions [186]. While RBF networks excel at modeling complex, nonlinear relationships, their performance can degrade with imbalanced datasets unless class weighting with the snowball method proposed [187] and the multi-font character recognition to improve the accuracy of the minor class [188]. Overfitting is also common if too many basis functions are used, necessitating the use of regularization methods, such as Least Squares Regulization (Ridge Regression) ensures numerical stability and smoother decision boundaries [189]. Another promising solution the MLP networks can manage high-dimensional data but require careful architecture tuning and regularization in to overcome this difficulty. In [190], a high-dimensional time-series classification model (HDTSCM) based on an MLP and moving average mode is used. MLPs excel at capturing complex, nonlinear relationships but require adjustments to their loss functions (e.g., focal loss, weighted loss) to address class imbalance and are prone to overfitting without techniques such as dropout or batch normalization [191,192,193]. DT handles high-dimensional data but can become overly complex and overfit, requiring pruning to improve generalization [194]. They are biased toward the majority class and need resampling techniques to address class imbalance [195]. Although decision trees can capture some nonlinearity, they are highly sensitive to noise [196]. RD improve upon decision trees by handling high-dimensional data more effectively and using feature importance to reduce dimensionality [197]. RFs are less prone to overfitting due to bagging and can capture complex relationships, although they may still require resampling techniques or balanced Random Forests to address class imbalance [198]. So far, advance and novel DLM strategies (CNN, RNN, Transformer architectures) offer the best solution for addressing the cited challenge. It excels at handling large feature maps from images or spectrograms, time-series analysis and can effectively model complex, nonlinear patterns [199]. Data augmentation techniques help mitigate class imbalance, while dropout improves generalization [200,201].

A closer examination of Table 4 highlights the importance of each step in the CM process for addressing data-driven challenges and building effective models. As discussed, the process begins with data collection, where different data types have shown varying effectiveness depending on the fault type in IMs. Using multimodal data can further improve performance and enhance the generalization of CM systems, making them less specialized in detecting only a single fault type. The next stage involves data cleaning and signal processing, where diverse techniques have been employed; recent trends indicate a reduction in preprocessing steps, making intelligent systems more practical for industrial deployment with minimal resource usage. For fault classification, two categories of approaches emerge: earlier methods relying on explicit feature extraction, which perform well in specific cases and remain attractive for embedded systems with limited resources; and more advanced methods, including CNNs, RNNs, and Transformers, which require greater computational power but can automatically extract features from raw data, making them well-suited for cloud- and IoT-based CM implementations.

## 7. Conclusions

This paper highlighted the general process of a fault detection system for IMs and provided an insightful review of several steps and approaches that can be used to design an efficient diagnosis system for IMs.

Each approach is evaluated by highlighting its weaknesses and strengths in solving data classification problems. In this study, we reviewed the common faults that affect the proper operating conditions of IMs and examined how each fault typically manifests. In other words, we defined the concept of fault signatures and discussed how understanding these signatures can support the development of an effective and reliable CM system. Subsequently, we outlined the various strategies used to obtain or generate data for condition monitoring. These strategies can be broadly categorized into two main approaches: the first involves collecting real-world data from physical systems, while the second relies on generating synthetic data through modeling and simulation techniques. Several parameters can be used to characterize faults in IMs, and these parameters may originate from various domains such as electrical, mechanical, thermal, and others. Although recent trends in CM increasingly aim to eliminate the need for manual signal preprocessing by leveraging deep learning techniques, signal processing remains a crucial step in many applications. Traditional methods such as the FFT and its variants are still widely employed in recent studies. Additionally, techniques like WT and HT continue to play a significant role in feature extraction and noise reduction. Recently, ML and DLM techniques have emerged as powerful tools for identifying fault patterns and signatures, as they exploit the statistical characteristics of the available data. Various ML and DLM strategies have been used for final CM process step which the data calssification. In our review, we focused primarily on supervised learning techniques, selecting two main branches: ANNs and ensemble methods based on decision trees. The existing categorization in data-driven engineering, which divides intelligent systems into ML- and DL-based approaches, remains valid when applied to IM CM. ML methods such as SVM, DT, RF, and MLP perform effectively for detecting pronounced faults but struggle with early-stage faults; nevertheless, they require relatively fewer computational resources. In contrast, deep learning models (CNNs, RNNs, and Transformers) simplify the CM process by reducing the need for extensive preprocessing, though they demand higher computational power. In summary, traditional ML methods are well-suited for environments that do not require very high accuracy and for embedded systems with limited resources, whereas advanced DL approaches are recommended for sensitive industrial applications, such as nuclear installations, where downtime could result in severe consequences.

## Figures and Tables

**Figure 1 sensors-25-05942-f001:**
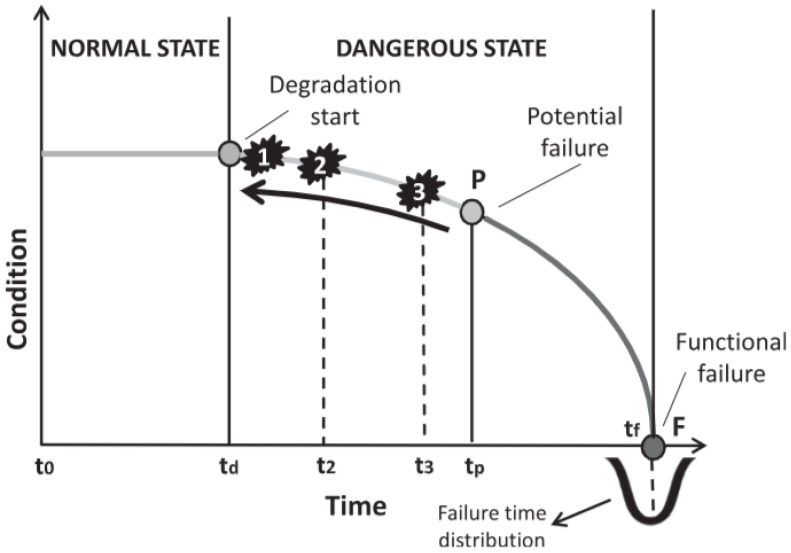
Predictive maintenance in the context of the P–F curve [6].

**Figure 2 sensors-25-05942-f002:**
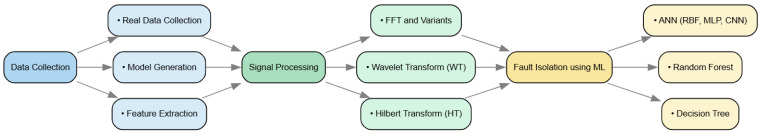
Overall Process of Condition-Based Monitoring for Induction Motors.

**Figure 3 sensors-25-05942-f003:**
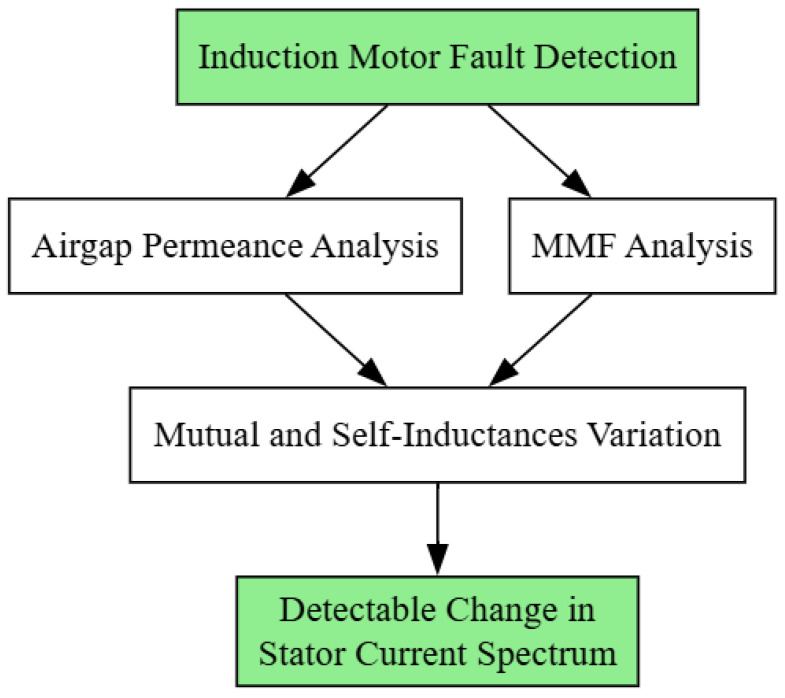
Induction motor fault detection from airgap and MMF analysis to stator current changes.

**Figure 4 sensors-25-05942-f004:**
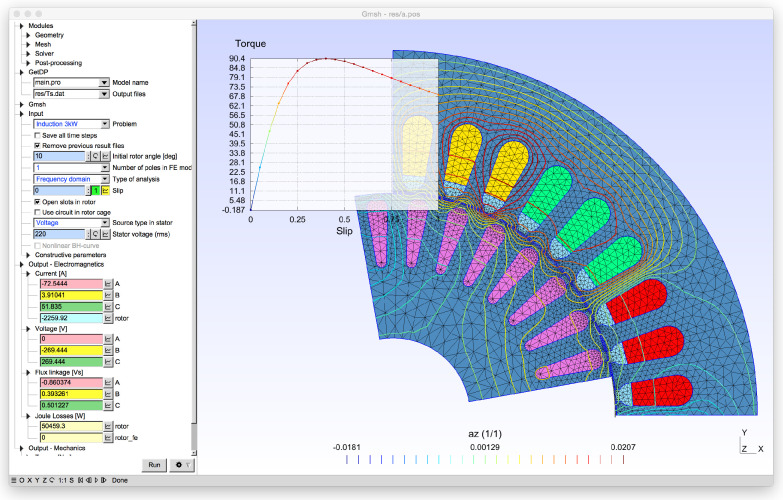
FEA -based IM model using GMSH/GetDp application [50].

**Figure 5 sensors-25-05942-f005:**
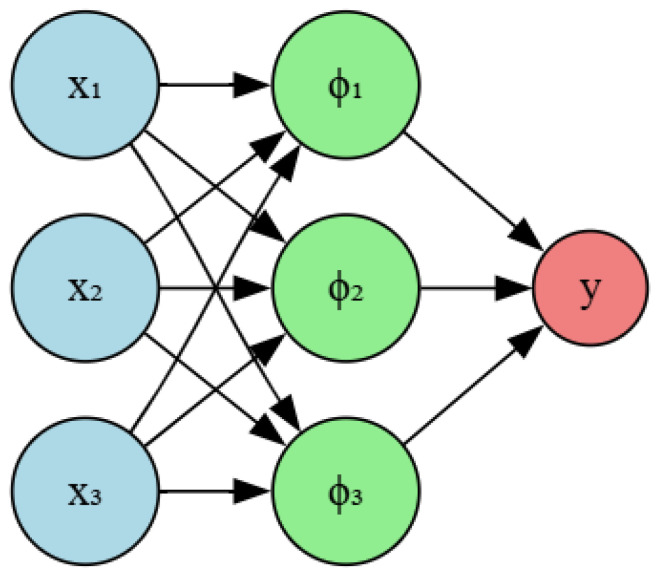
Architecture of a RBF network. The network consists of an input layer, a hidden layer with radial basis functions, and an output layer.

**Figure 6 sensors-25-05942-f006:**
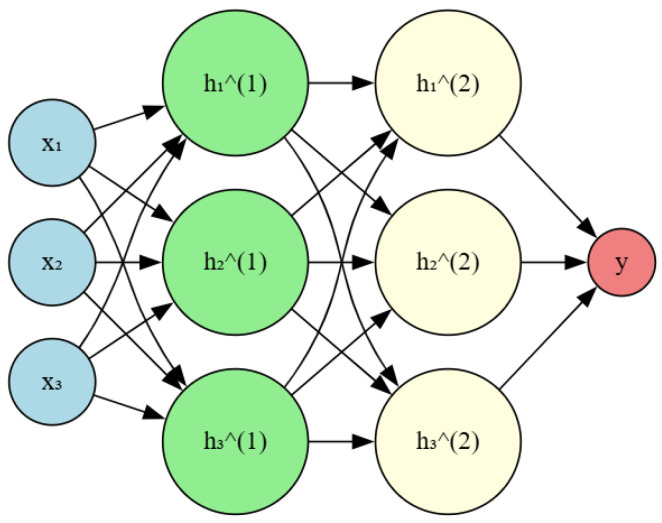
Architecture of a MLP network. The network consists of an input layer, two hidden layers, and an output layer.

**Figure 7 sensors-25-05942-f007:**
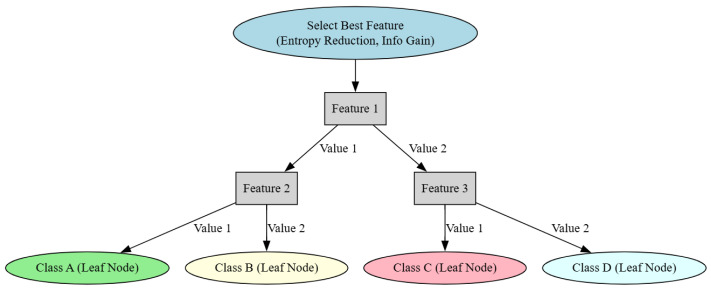
Workflow of the DT Algorithm.

**Figure 8 sensors-25-05942-f008:**
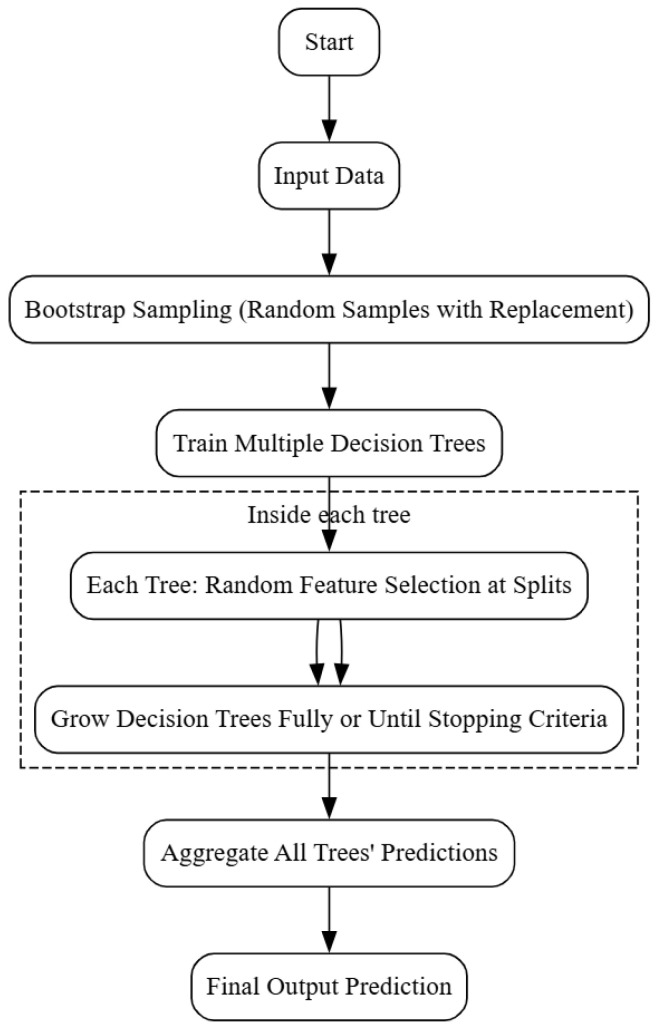
Workflow of the Random Forest Algorithm.

**Figure 10 sensors-25-05942-f010:**
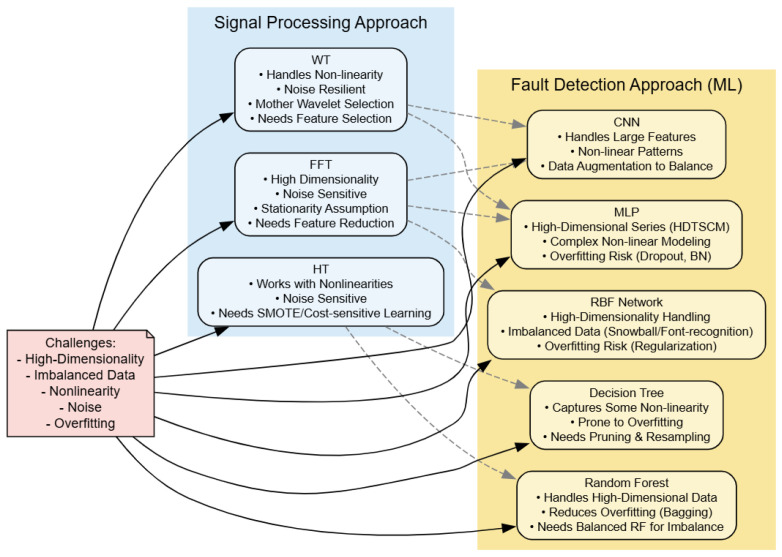
Challenges and efficient methods for induction motor fault detection: A post-review synthesis.

**Table 1 sensors-25-05942-t001:** Classification of faults, their detection methods, and descriptions.

Name of Fault	Classification	Description	References
Bearing failure	Mechanical	A prevalent origin of vibration (2–60 kHz). Thermal imaging is more effective for detection.	[14,15,16]
Broken rotor bars	Mechanical	Low amplitude makes detection challenging. Current signals provide higher sensitivity.	[17,18]
Misalignment	Mechanical	Detected via infrared thermography (IRT) and vibration signature analysis.	[19,20]
Rotor mass unbalance	Mechanical	Centrifugal force induces heightened vibration in the rotor and stator.	[21,22,23]
Air gap eccentricity	Mechanical	Spectral analysis of apparent power modulus detects early-stage faults.	[24,25]
Coil and lamination defects	Electrical	Requires reduction of eddy current losses.	[26,27]
Stator winding failure	Electrical	Characterized by current anomalies. Thermal imaging is applicable.	[17,28]
Crawling	Electrical	Harmonic distortion variations in the Larke plane axes are detectable.	[29]
Unbalanced supply voltage/current	Electrical	Dynamic symbolic state machines (DSSMs) identify voltage imbalances.	[17]
Single Phasing	Electrical	Infrared examinations facilitate fault identification.	[27]
Earth fault	Electrical	High-resistance grounding manages fault currents effectively.	[27,30]
Ambient temperature	Environmental	May lead to inaccurate measurements.	[27,30,31]
Contamination	Environmental	Particulate matter in metallic construction exceeds final packing by 200%.	[27,31]
Humidity	Environmental	Negligible impact on temperature measurement.	[30]

**Table 2 sensors-25-05942-t002:** The main parameters of data acquisition and their associated indication [33].

Parameters	Indication	Examples/Insights
Current (I)	The current flowing through the motor windings reflects operational status and health.	Stator winding faults: Increased current due to insulation failure or short-circuits. Rotor bar damage: Asymmetrical current or spikes indicate broken rotor bars. Overload conditions: Excessive current from higher-than-rated load. Unbalanced phases: Imbalance in three-phase motors suggests supply or winding issues.
Voltage (V)	Voltage affects motor performance and efficiency.	Voltage imbalances: Cause uneven torque and mechanical stress. Undervoltage/overvoltage: Indicates supply issues or improperly sized motor. Harmonics/sags: Distortions reduce motor lifespan and performance.
Power (P)	Power consumption reflects load conditions and efficiency.	Overloading: Increased power due to excessive mechanical load. Rotor bar faults: Inconsistent power under load indicates rotor damage. Inefficiencies: High power vs. output suggests friction/misalignment.
Speed (N)	Rotational speed deviations detect abnormalities.	Mechanical faults: Slowdowns indicate bearing wear or misalignment. Electrical faults: Speed reductions from rotor/stator issues. Overheating: Speed reduction due to overloaded motor.
Temperature (T)	Temperature monitoring identifies overheating or insulation issues.	Overheating: From excessive load or poor ventilation. Insulation deterioration: High temperatures degrade insulation. Bearing failure: Heat from mechanical friction.
Vibration (Vib)	Vibration analysis detects mechanical faults.	Rotor imbalance: Excessive vibrations from imbalance/misalignment. Bearing wear: Abnormal vibrations from worn bearings. Misalignment: Motor-load misalignment causes vibrations.

**Table 3 sensors-25-05942-t003:** Statistical survey related to fault detection in IM using current analysis and Vibration Analysis.

Publisher	MCSA	Vibration Analysis
IEEE	277	516
Elsevier	299	1781

**Table 4 sensors-25-05942-t004:** A summary of deep learning approaches for fault diagnosis.

Reference	Type of Data Used	Signal Processing	Model Application	Performance Metrics
RF				
Abdulkareem et al. (2025) [173]	Vibration and Temperature	Imputation, Z-score, FT Features, Min–Max Normalization	Bearing fault and Load imbalance	Among all models, Random Forest delivered the strongest results, attaining an accuracy of 91%.
Patel et al. (2016) [174]	Vibration Signal	Statistical Features	Bearing fault detection	A further analysis with the four most important features led to 100% prediction success.
Pohakar (2025) [175]	Torque, Speed, Currents, Power	Imputation, Z-score/IQR, Min-Max Normalization, PCA, and FT Features	7 Universal Categories of IM faults	The RF model reached 93.1% accuracy, highlighting its dependable and steady performance in motor fault detection across categories.
MLP				
Ghate (2010) [123]	Stator Current	Statistical Parameters are used as input feature space and PCA	Stator winding inter-turn short and rotor dynamic eccentricity	The reduced MLP NN achieved low MSE (0.046 test, 0.030 CV) with high accuracy (98.25% test, 96.22% CV) for fault diagnosis.
Jin (2025) [176]	Operational Data (Vibration, Temperature, Speed)	Continuous Wavelet Transform (CWT) features integrated with a dynamic multi-head attention mechanism	RUL prediction and anomaly detection	The proposed approach enhances the accuracy and reliability of RUL predictions, supporting more effective predictive maintenance in industrial settings.
Santos (2021) [177]	Acoustic Signals	MFCCs; Noise reduction	Incipient fault classification	Experiments with varying loads and voltage unbalance, typical of industrial settings, achieved over 97% accuracy
Convolutional Neural Networks (CNNs)				
Nazemi (2024) [178]	Three-phase Stator Currents	A current to image transformation mechanism	Stator inter-turn faults	The proposed CNN effectively detects SITFs with superior accuracy and online potential.
Abdelmaksoud (2023) [179]	Image Data (Voltages, Currents, Torque, and Speed)	d-q Lissajous imaging with single multi-channel inputs	Locked-rotor, overload, voltage unbalance, overvoltage, and undervoltage	Strong cross-machine generalization; open-set and source-free adaptation.
Lee (2019) [180]	Vibration Signal	Raw data	Rotor fault and bearing fault	Accuracies of 98% for normal operation, 98% for rotor faults, and 100% for bearing faults in IM.
Transformer Models				
Chen (2023) [181]	Stator Current	Current signals into time-domain images using the Instantaneous Square Current Value (ISCV)	Bearing fault	Average diagnostic accuracies of 96.60% (PU dataset) and 94.87% (SZTU dataset) using the ISCV-ViT model.
Ali (2025) [182]	Multivariate Time-series	Combining Transformer feature extraction with DNN classification for fault detection	Binary and multi-class detection of IMs faults (mechanical and electrical)	Binary accuracies of 99.97% (TMFD) and 98.26% (MFD), for multi-class 99.97% (TMFD) and 98.39% (MFD).
Choi (2025) [183]	Multidimensional Power Quality Data (Voltage, Current, and Harmonics)	Multi-scale feature analysis, frequency gating, and SHAP-based interpretation.	IM’s shaft unbalance, bearing and stator winding faults	Achieved 99.9% accuracy with 0.1% false alarm rate and 0.2% missed detection rate
RNNs				
Vos (2022) [184]	Vibration Signals	A two-step LSTM configuration and statistical feature	Bearing anomaly detection	LSTM2-OCSVM architecture improved sensitivity to bearing.
Ahsan (2025) [185]	Vibration Signals under three load conditions (100 W, 200 W, 300 W)	1D vibration signals were transformed into 2D time–frequency images using CWT	Different bearing fault	Achieved 100% training accuracy and validation accuracies of 96.43% (100 W), 97.47% (200 W), and 95.06% (300 W).

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
