# Peer review of "Advancements in Induction Motor Fault Diagnosis and Condition Monitoring: A Comprehensive Review"

_sensors, 2025, doi:10.3390/s25195942_

Round 1

Reviewer 1 Report

Comments and Suggestions for Authors 1. The references in this paper include a large number of studies published between 1990 and 2010. While a review paper needs to introduce various methods at the beginning, overly outdated references should be briefly covered with a reduced proportion—their reference value for current research is quite limited. Personally, it is recommended to include more reviews of technologies from the past decade. 2. This paper also cites a large number of conference papers. However, many methods presented in conference papers are immature and thus have not been widely adopted. The authors should focus primarily on high-quality journal papers, with the number of journal papers exceeding that of conference papers. 3. The paper lacks the authors’ own perspectives and is merely a compilation and summary of existing references. 4. Figures 6 to 8 are meaningless. Instead, they should display the number of published papers corresponding to methods based on different principles. 5. Based on their own research and engineering experience, the authors should provide evaluations of different fault types and methods. Currently, the paper only lists them without any assessment. 6. Even after reading the entire paper, I still cannot determine which method is most appropriate for a specific type of fault. 7. Many methods are only briefly introduced, with a lack of in-depth analysis.

Reviewer 2 Report

Comments and Suggestions for Authors

The authors carried out a literature review in Induction Motors’ faults detection, highlighting areas of deficiency and obstacles.  The main steps of diagnosis process are discussed, presenting existing approaches for each step.

The effectiveness of each fault classification approach is discussed next, in addressing data-driven challenges such as high-dimensionality, class imbalance, non-linearity, noise and overfitting.

The rising transition to data-driven strategies is observed, with deep learning increasingly employed to attack complex challenges of fault diagnosis.

Future research is shown to shift into intelligent, real-time condition monitoring systems.

The authors are advised to address the following issues in a carefully revised version of the manuscript:

Line 67: better:  “harsh operating conditions”…

Line 69: lack of meaning in sentence – no end.

Figure 3: Below each specific fault type at right, it is supposed to be stated the most common diagnostic tests. However, this is not consistent with all fault types presented.

Is there a standardized induction motors fault classification to use as reference (IEEE, NEMA, etc)?

Section 3, which is the most important part of the diagnostic process, is presented in a superficial manner.

On the other hand, Section 4 which is a rough quantitative search, is given a not-deserved gravity in the manuscript.

Figure 4 caption: something goes wrong here please correct.

Figure 5: not acceptable, increase resolution. Add description, what variable is modeled (electric field etc), indicate measurement scale.

Line 209: please remove overstatement.

Line 234-235: This does not appear to be a similar evolution.

Section 5 needs further discussion.

Section 7 needs the main indicators to be mathematically stated and further discussed.

General remark: The review lacks in coherence. The comparative discussion is not in depth. The Conclusions are simplistic.

If the authors really want to present a valid review, they need to significantly enhance their effort with the manuscript.

Comments on the Quality of English Language

English need further improvement,correction of typos and faults.

Round 2

Reviewer 1 Report

Comments and Suggestions for Authors

The author has carefully responded to the previous comments and made corresponding revisions.

Reviewer 2 Report

Comments and Suggestions for Authors

The authors made significant improvements in their revision.

The revised manuscript can be accepted.